# Adherence to Cysteamine Therapy Among Patients Diagnosed with Cystinosis in Saudi Arabia: A Prospective Cohort Study

**DOI:** 10.3390/pharmacy12040123

**Published:** 2024-08-08

**Authors:** Reem Algasem, Nedaa Zainy, Essam Alsabban, Hamad Almojalli, Khalid Alhasan, Tariq Ali, Deiter Broering, Hassan Aleid

**Affiliations:** 1Department of Pharmacy, King Faisal Specialist Hospital and Research Centre, Riyadh P.O. Box 3354, Saudi Arabia; 2Department of Pediatric Nephrology, King Faisal Specialist Hospital and Research Centre, Riyadh P.O. Box 3354, Saudi Arabia; 3Department of Kidney and Pancreas Transplant, King Faisal Specialist Hospital and Research Centre, Riyadh P.O. Box 3354, Saudi Arabia; halmojalli@kfshrc.edu.sa (H.A.); ali99@kfshrc.edu.sa (T.A.);

**Keywords:** cystinosis, cysteamine, treatment adherence, Saudi Arabia

## Abstract

Cystinosis is a rare autosomal recessive disorder in which cystine crystals accumulate within the cellular lysosomes, causing damage to multiple organs. Due to challenges with the stringent cysteamine treatment regimen and side effects, adherence is often sub-optimal. This study aimed to assess the level of adherence to cysteamine therapy among cystinosis patients in Saudi Arabia and its impact on their quality of life. Electronic medical record data of 39 cystinosis patients from the Department of Nephrology at King Faisal Specialist Hospital and Research Center in Saudi Arabia were reviewed, and 25 patients were included in this study. Out of the 25 patients included in the final analysis, 64% (*n* = 16) were female. The mean age was 19.04 years. Almost all patients (23/25, 92%) were on oral IR cysteamine therapy, and 52% (13/25) were on topical cysteamine eye drop treatment. Of the 15 patients who responded to the Morisky Medication Adherence Scale-8 (MMAS-8) questionnaire, only 4 (26.7%) were highly adherent to cysteamine therapy. Most of the respondents (7/15, 46.7%) showed a medium level of treatment adherence. Based on the medication possession ratio for oral cysteamine, only 6 out of 23 patients (26.1%) were found to be 96–100% adherent. For the cysteamine eye drops, only 5/13 patients (38.4%) were 76–95% adherent. The 36-Item Short Form Health Survey (SF-36) used to assess patients’ health-related outcomes showed that their quality of life was affected in the domains of ‘social functioning’ and ‘energy/fatigue.’ Despite a small sample size, this study shows sub-optimal adherence to cysteamine treatment in patients from Saudi Arabia. The possible reasons for low treatment adherence could be a high frequency of administration and treatment-related side effects.

## 1. Introduction

Cystinosis is a rare autosomal recessive disorder in which cystine crystals accumulate within the cellular lysosomes, resulting in damage to multiple organs [1]. It is caused by mutations in the cystinosin-encoding CTNS gene that disrupt the transportation of cystine out of the lysosomes. Cystine accumulation within the lysosomes is responsible for the clinical manifestations of cystinosis [2]. The primary clinical presentations of cystinosis include infantile nephropathic cystinosis, late-onset nephropathic cystinosis, and non-nephropathic ocular cystinosis [3]. Infantile nephropathic cystinosis is the most frequently diagnosed and the most severe form of cystinosis, manifesting as renal Fanconi syndrome that progresses to end-stage renal disease (ESRD) at a young age. Typically, the first extra-renal pathological finding in cystinosis patients is in the cornea in the form of cystine crystals [3]. Moreover, cystinosis is a systemic disorder, and organ systems like the endocrine, muscular, and central nervous systems are also affected, besides the kidneys and eyes.

The mainstay treatment for cystinosis is cysteamine, a cystine-depleting agent that has helped improve overall outcomes and life expectancy by delaying extra-renal manifestations and the progression to ESRD [1,4,5,6]. Cysteamine allows cystine to exit the lysosomes via transporter mechanisms other than cystinosin [2]. Although, unlike many other orphan diseases, treatment is available for cystinosis, its effectiveness relies on the extent of adherence, which has often been found to be sub-optimal, primarily due to challenges with the treatment regimen and gastrointestinal and other side effects like halitosis and sulfuric body odor [2,4,5]. Moreover, cysteamine treatment is not curative and must be initiated early and taken lifelong to ensure its therapeutic benefits are entirely utilized.

Cysteamine is available in immediate-release (IR) and delayed-release (DR) formulations. The IR formulation of cysteamine (CYSTAGON^®^), available in 50 mg and 150 mg capsules, has been approved for use in the US and Europe since 1994 and 1997, respectively [4,7,8]. The recommended oral dose of IR cysteamine in children up to 12 years is 1.3 g/m^2^/day (or 60 mg/kg/day) divided into four doses and increased gradually over four to six weeks. The maximum recommended dose is 1.95 g/m^2^/day or 90 mg/kg/day [7]. Approved by the FDA in 2013, the DR formulation PROCYSBI^®^ was developed to address the frequent dosing challenges associated with the IR formulation. PROCYSBI^®^ is available in 25 and 75 mg capsules, and the recommended dose is 1.3 g/m^2^/day in divided doses every 12 h [4,9]. However, the side effect profiles of the IR and DR cysteamine formulations are comparable and primarily include vomiting, abdominal pain, diarrhea, loss of appetite, drowsiness, and skin rash.

Corneal manifestations of cystinosis include photophobia, blepharospasm, corneal erosions, superficial punctate keratopathy, and band keratopathy. However, as the cornea is avascular, oral cysteamine fails to have any impact on corneal outcomes [10]. Topical application was found to be better suited for this purpose. Hence, an ophthalmic formulation, CYSTARAN^®^, was approved by the FDA in 2012 with the recommendation to administer one drop in each eye every waking hour [11]. A significant challenge with topically applied cysteamine is that it is rapidly cleared from the eye due to its inherent defense mechanisms, like reflex tearing, frequent blinking, and nasolacrimal drainage [12]. Thus, frequent administration of the ophthalmic cysteamine becomes inevitable to ensure efficient removal of corneal cystine crystals. Also, CYSTARAN^®^ has stringent storage requirements; it must be stored in a freezer (−25 °C to −15 °C) and thawed for 24 h before use. The thawed ophthalmic solution must be used within 7 days [11]. Efforts to tweak the aqueous ophthalmic formulation, enabling more favorable storage requirements and less frequent administration, has led to the development of CYSTADROPS^®^, a more viscous formulation that was approved by the FDA in 2020 [13].

An open-label, phase 3, two-arm, multi-center RCT in France on cystinosis patients (≥2 years) by Liang et al. compared the efficacy of the viscous 0.55% cysteamine hydrochloride ophthalmic formulation with the standard 0.10% cysteamine hydrochloride ophthalmic formulation. The trial results showed that the viscous cysteamine eye drops effectively reduced the corneal crystal density, photophobia, corneal cystine crystal scores (CCCS), and corneal crystal depth [10]. 

Despite an improvement in the prognosis of cystinosis over the years due to the availability of oral and topical cysteamine treatments, early initiation of these treatments and a high level of adherence are critical to delaying disease progression [12]. Stringent adherence to treatment is also essential to ensure long-term organ protection and an improvement in QoL. Ariceta et al. have evaluated the adherence to CYSTAGON^®^ treatment in 34 cystinosis patients through a voluntary, anonymous survey. Treatment was administered by mothers (100%) or fathers (83%) to children less than 11 years old. However, less parental involvement was seen for patients over 11 years of age, and the treatment was primarily self-administered. The study revealed that, while 94% of the patients under 11 adhered to the CYSTAGON^®^ treatment as prescribed, only 50% of the older patients demonstrated a high level of adherence, with the rest reporting frequent schedule changes, missed doses, never using recommended doses, and not caring about the drug prescription. This study identified a decline in the treatment adherence level in teenagers and adults as a significant concern [14].

The aim of this study was to assess the level of adherence to cysteamine therapy among cystinosis patients in Saudi Arabia and its impact on the QoL. Through this study, we would like to evaluate the relationship between adherence to cysteamine therapy and the overall well-being of cystinosis patients in the Saudi Arabian context. 

## 2. Materials and Methods

### 2.1. Study Design and Population 

This prospective cohort study was conducted at the Department of Nephrology at King Faisal Specialist Hospital and Research Center (KFSH&RC) in Riyadh, Saudi Arabia. Patients of all ages, male or female, with an established diagnosis of cystinosis and receiving cysteamine therapy were included in the study. 

### 2.2. Data Collection 

This study was approved by the Office of Research Affairs at KFSH&RC. Electronic Medical Files of patients with cystinosis who had visited the Nephrology Clinic at KFSH&RC were reviewed. Demographic data of patients, details related to cysteamine treatment (starting date, type of treatment, dosage, and frequency), and information about any other therapeutic medications the patients were taking were systematically collected. Patients or their parents underwent telephonic interviews, during which we also obtained informed consent for their participation in this study. Adherence to cysteamine therapy (oral therapy and eye drops) was assessed using two measures, Morisky Medication Adherence Scale-8 (MMAS-8) and Medication Possession Ratio (MPR). 

The MMAS-8 is a structured, self-reported measure that evaluates the level of treatment adherence. The scale ranges from 0 to 8, where a score of 8 indicates high adherence, 6 to 8 reflects medium adherence, and a score below 6 suggests low adherence. The scale was analyzed using the Morisky Widget software, Chicago, IL, USA [15].

The second measure was medication possession ratio (MPR) that measures the percentage of time a patient has access to medication through the following formula [16]:(1)MPR=Total day's supply in periodLast fill date−first fill date+last fill days' supply

MPR of less than 50% indicates low adherence, 50–75% indicates moderate adherence, 76–95% indicates good adherence, and 96–100% indicates excellent adherence.

We also evaluated patients’ health-related QoL using the 36-Item Short Form Health Survey (SF-36). It is a self-reported survey that measures the health outcome using eight main domains: physical functioning, bodily pain, role limitations due to physical health, role limitations due to personal or emotional problems, emotional well-being, social functioning, energy/fatigue, and general health perceptions [17].

### 2.3. Statistical Analysis

Continuous variables were presented as mean or median as appropriate based on the normality of the data, and categorical data were presented as proportions. Descriptive data on adherence to oral cysteamine tablets and topical eye drops were analyzed based on the MPR calculations and responses to the MMAS-8. The overall QoL evaluation was derived from the SF-36 responses.

## 3. Results

### 3.1. Patient Recruitment and Baseline Characteristics

The patient recruitment scheme is depicted in Figure 1. Twenty-five cystinosis patients were included in the final analysis. The baseline characteristics of the study participants are summarized in Table 1. The mean age of the patients was 19.04 years (standard deviation [SD] = 6.78), and the majority of patients were female (64%). Barring 2 patients, the remaining 23 (92%) were prescribed oral IR cysteamine therapy, and only about half (13/25; 52%) were prescribed cysteamine eye drops.

### 3.2. Evaluating the Level of Adherence: MMAS-8 Responses

Of the 25 patients, 15 completed and responded to the MMAS-8 questionnaires. Their responses and corresponding adherence scores are listed in Table 2. The MMAS-8 score analysis revealed that, among the 15 respondents, only 4 (26.7%) demonstrated high adherence to cysteamine therapy, achieving a score of 8. Also, four respondents (26.7%) were categorized as having low adherence with a score < 6. Most of the respondents (*n* = 7; 46.7%) exhibited a medium level of adherence, achieving scores ranging between 6 and 7.

### 3.3. Evaluating the Level of Adherence: MPR Analysis

The adherence to cysteamine therapy was additionally assessed through the MPR (Table 3). For oral cysteamine therapy, 6/23 patients (26.1%) were found to be highly adherent to oral cysteamine treatment, showing an MPR of 96–100%, followed by 5/23 patients (21.7%) with an MPR of 76–95%. However, 21.7% of the patients (5/23) were also found to exhibit a low adherence to oral cysteamine therapy, with MPR less than 50%.

Interestingly, none of the 13 patients prescribed cysteamine eye drops fit the excellent adherence category (MPR of 96–100%) for the topical treatment. Although most patients (5/13; 38.4%) exhibited good adherence (MPR of 76–95%), about 30.8% of the patients (4/13) were found to show only moderate or low levels of adherence to the ophthalmic cysteamine treatment with an MPR ranging between <50 and 75%.

Also, a total of 8 patients from both the oral and ophthalmic cysteamine treatment groups were found to have an MPR exceeding 100% due to overfilling of prescriptions, with 5 patients and 3 patients from the 2 groups filling once per year.

### 3.4. Evaluating the Health-Related QoL: SF-36 Responses

Twenty-two participants of the twenty-five responded to the SF-36 questionnaire. Their responses for the 36 items on the SF-36 were analyzed, and the mean score for each of the eight domains that impact patients’ health-related QoL was computed (Table 4) [17,18].

The mean (±SD) SF-36 scores for the eight domains varied, ranging from 59.66 (±24.38) for ‘social functioning’ to 82.55 (±20.6) for ‘emotional well-being’. The two domains from the SF-36 survey that were the most impacted among the cystinosis patients were ‘social functioning’ and ‘energy/fatigue’. Within these domains, the key aspects affected were ‘physical health or emotional problems interfering with normal social activities’ and ‘feeling tired and worn out’. Most patients reported experiencing either extreme or frequent interference with their social activities due to physical health concerns or emotional problems. Most patients also reported feeling tired or worn out all the time.

## 4. Discussion

The availability of cysteamine since the early 1990s has helped improve the prognosis of cystinosis patients by delaying renal and extra-renal complications and prolonging life expectancy. It is available in two formulations: IR and DR.

Langman et al. conducted an open-label, crossover, randomized controlled trial (RCT) to compare the efficacy of the IR and DR formulations. This trial included 43 patients (*n* = 38 for primary efficacy analysis) with a mean age of 12. It showed that the DR formulation was non-inferior to the IR formulation in terms of efficacy, assessed by measuring the WBC cystine levels at the end of each 3-week crossover period [19]. Moreover, to ensure that the efficacy of the DR formulation is sustained in the long term, Langman et al. extended their study for an additional 2 years. Forty of the forty-one patients who had completed the short-term study opted to enroll in the long-term study. The twice-daily administered DR formulation improved quality of life (QoL) parameters and maintained optimal mean WBC cystine levels. The QoL parameters were assessed using the Pediatric Quality of Life Inventory (PedsQL) questionnaire. A significant change in patients’ social, school, and total functioning was observed immediately after switching from IR to DR formulation, and this improvement was sustained even after 2 years of DR therapy [20].

In a more recent study by van Stein et al., the investigators retrospectively compared the cystine and cysteamine levels in 17 patients after a single dose of IR and DR cysteamine formulations. They found that the DR formulation was as effective as the IR formulation, albeit with fewer side effects. While 88.2% (15/17) of patients experienced gastrointestinal (GI) side effects like nausea, vomiting, abdominal pain, and sulfurous body odor with the IR cysteamine, only 35.3% (6/17) experienced such GI effects, and 29.4% (5/17) of patients experienced sulfurous body odor with the DR cysteamine. Moreover, four of the six patients described the GI side effects to be less severe than what they had experienced with the IR formulation, and three of the five patients described the sulfurous body odor to be less intense with the DR formulation than the IR formulation [21].

Cysteamine treatment, however, is not curative and must be taken lifelong to ensure organ preservation through the continuous depletion of cystine crystals from cells [22]. Stringent adherence to cysteamine treatment is crucial to optimize outcomes and delay disease progression. Non-compliance with the cysteamine treatment jeopardizes its efficacy and compromises the QoL of cystinosis patients. Moreover, treatment non-adherence and the resulting inevitable need for subsequent interventions like kidney transplants impose a financial burden on cystinosis patients as well as the healthcare system [23].

This study shows that sub-optimal adherence to the cysteamine treatment, as assessed by patient-reported MMAS-8 and pharmacy-related MPR measures, is a significant challenge in the management of cystinosis. The evaluation of patient QoL using the SF-36 tool demonstrated that, despite being under treatment, physical health challenges and emotional issues patients face with cystinosis interfere with their ability to engage in everyday social interactions and activities. The patients also reported feeling tired and worn out consistently, underscoring an urgent need to identify the specific problem areas that contribute to non-adherence and design therapeutic management strategies that can address these challenges.

The relatively small sample size used in this study is a limitation. Also, an assessment of intra-leukocyte cystine levels, the current gold standard for monitoring therapy adequacy, was unavailable to us, representing another study limitation [24]. Future studies incorporating this assessment can provide additional insights and more objective evaluations of the extent of cysteamine treatment non-adherence. Adherence assessment used in this study complements cystine testing, as it captures patients’ attitude to treatment beyond the time of cystine testing. Thus, despite the limitations, we believe this study can serve as a foundation for designing larger prospective studies to better understand the precise reasons for non-adherence to cysteamine treatment within the local Saudi population and the Gulf population as a whole.

We would also like to acknowledge the challenges we encountered in patient follow-up. As this study was conducted during the COVID-19 pandemic, we could not conduct face-to-face interviews with the study participants and had to rely exclusively on telephonic interviews. This may have potentially led to some communication gaps and incomplete information sharing by the study participants.

## 5. Conclusions

This study demonstrates sub-optimal adherence to cysteamine treatment in cystinosis patients from Saudi Arabia. It emphasizes the need to conduct larger prospective studies that incorporate intra-leukocyte cystine level measurements to monitor treatment efficacy as an indirect measure of treatment adherence. Such studies can further elucidate the factors contributing to non-adherence.

## Figures and Tables

**Figure 1 pharmacy-12-00123-f001:**
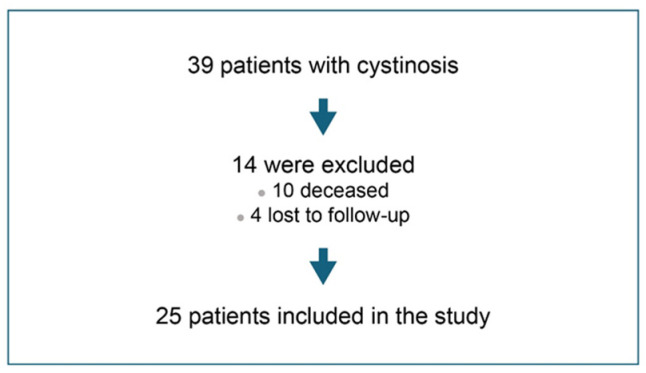
Patient recruitment scheme.

**Table 1 pharmacy-12-00123-t001:** Baseline characteristics of the participants (*n* = 25) of the study.

	Mean + SD *
Age (years)	19.04 + 6.78
Weights (Kg)	41.3 + 14.09
Height (cm)	137.74 + 18.12
	N (%)
**Gender**	
Male	9 (36)
Female	16 (64)
**Cysteamine therapy**	
Oral (IR)	23 (92)
Eye drops	13 (52)

* SD—Standard Deviation.

**Table 2 pharmacy-12-00123-t002:** Results of the Morisky Medication Adherence Scale-8 (MMAS-8) for each patient.

Patient No.	Item 1	Item 2	Item 3	Item 4	Item 5	Item 6	Item 7	Item 8	Overall
1	No (1)	No (1)	No (1)	No (1)	Yes (1)	No (1)	No (1)	Never/rarely (1)	8
2	No (1)	No (1)	No (1)	No (1)	Yes (1)	No (1)	Yes (0)	Never/rarely (1)	7
3	No (1)	No (1)	No (1)	Yes (0)	Yes (1)	No (1)	No (1)	Never/rarely (1)	7
4	No (1)	No (1)	No (1)	No (1)	Yes (1)	No (1)	Yes (0)	Never/rarely (1)	7
5	No (1)	No (1)	No (1)	No (1)	Yes (1)	No (1)	Yes (0)	Never/rarely (1)	7
6	Yes (0)	Yes (0)	No (1)	No (1)	Yes (1)	No (1)	Yes (0)	Sometimes (0.5)	4.5
7	Yes (0)	No (1)	Yes (0)	Yes (0)	No (0)	No (1)	No (1)	Sometimes (0.5)	3.5
8	No (1)	No (1)	No (1)	No (1)	Yes (1)	No (1)	No (1)	Never/rarely (1)	8
9	No (1)	Yes (0)	No (1)	No (1)	No (0)	No (1)	No (1)	Never/rarely (1)	6
10	No (1)	Yes (0)	No (1)	No (1)	Yes (1)	No (1)	Yes (0)	Never/rarely (1)	6
11	No (1)	No (1)	No (1)	No (1)	Yes (1)	No (1)	No (1)	Never/rarely (1)	8
12	Yes (0)	No (1)	Yes (0)	No (1)	Yes (1)	Yes (0)	Yes (0)	Sometimes (0.5)	3.5
13	Yes (0)	No (1)	No (1)	No (1)	Yes (1)	Yes (0)	Yes (0)	Sometimes (0.5)	4.5
14	No (1)	No (1)	Yes (0)	No (1)	Yes (1)	No (1)	No (1)	Sometimes (0.5)	6.5
15	No (1)	No (1)	No (1)	No (1)	Yes (1)	No (1)	No (1)	Never/rarely (1)	8

**Table 3 pharmacy-12-00123-t003:** Assessment of adherence to cysteamine therapy using Medication Possession Ratio (MPR).

MPR	Oral Cysteamine (*n* = 23 Patients) N (%)	Cysteamine Eye Drops (*n* = 13 Patients) N (%)
Less than 50%	5 (21.7)	5 (21.7)
50 to 75%	3 (13)	3 (13)
76 to 95%	5 (21.7)	5 (21.7)
96 to 100%	6 (26.1)	6 (26.1)
More than 100% Overfill Filled once	4 (17.4) 0 (0)	4 (17.4) 0 (0)

**Table 4 pharmacy-12-00123-t004:** Health-related quality of life scores assessed by the 36-Item Short Form Health Survey (SF-36).

Scale	Mean + STD
Physical functioning	77.73 ± 32.39
Limitations due to physical health	71.59 ± 39.9
Limitations due to emotional problems	80.3 ± 39.39
Energy/fatigue	69.32 ± 24.89
Emotional well-being	82.55 ± 20.6
Social functioning	59.66 ± 24.38
Pain	75 ± 24.64
General health	74.55 + 24

## Data Availability

All data generated or analyzed during this study are included in this published article.

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
