# Peer review of "Adherence to Cysteamine Therapy Among Patients Diagnosed with Cystinosis in Saudi Arabia: A Prospective Cohort Study"

_pharmacy, 2024, doi:10.3390/pharmacy12040123_

Round 1

Reviewer 1 Report

Comments and Suggestions for Authors

In this study by Algasem Reem and colleagues, the authors have performed a prospective study on cystinosis patients aiming to assess the adherence to cyeasteamine therapy (both IR and delayed release). To this aim, the authors have used the Morisky Medication Adherence-Scale 8 (MMAS-8) questionnaire and the Medication Possession Ratio (MPR). Although the study has been well designed, there is in my opinion a serious limitation in the evaluation of adherence to therapy.

In particular, it is well recognized that measurement of intraleukocyte cystine levels represents the gold standard for monitoring therapy effectiveness and, inderectly, therapy compliance. We are aware that sometimes this test is not feasible due to the several criticisms that characterize this assessment. Moreover, a recent LC-MS/MS method for measurement of cysteamine plasma levels has been also proposed (DOI: 10.3390/ph17050649).   

However, in my opinion the authors should perform at least intraleukocytes cystine assessment. If it is not feasible, I would strongly reccomend to mention the importance of this test in the Discussion and Conclusion sections. 

Similarly, in the Conclusions the authors state that therapy adherence has been evaluated based on the tests performed in accordance to the study design. However, in my opinion, alongside with larger prospective studies, dosing intraleukocytes cystine should be considered as priority in order to assess both therapy efficacy and adherence.

Author Response

Please see the attachment, thank you!

Reviewer 2 Report

Comments and Suggestions for Authors

It would be interesting to know why half of the patients were not prescribed eye drops, although this manuscript outlines survey results and not individual reasoning.

Author Response

Please see the attachment, thank you!

Reviewer 3 Report

Comments and Suggestions for Authors

In this article by Reem Algasem and colleagues, the authors have conducted a prospective study aimed at evaluating the adherence of cystinosis patients to therapy with oral cysteamine (both immediate and delayed formulations) and cysteamine eye drops. In particular, the authors have adopted different approaches to evaluate patients' compliance. These include the Morisky Medication Adherence Scale-8 (MMAS-8) and the Medication Possession Ratio (MPR). Additionally, a Health Survey was conducted to rate and classify the quality of life (QoL) of each patient. From data collected, the authors conclude that sub-optimal adherence to cysteamine is a serious medical issue for these patients and require larger prospective studies to assess the main factors contributing to it. Moreover, emotional well-being and limitations due to emotional problems resulted as the principal domains affected by cysteamine therapy.

Despite the limited number of patients included in the analyses (n=25), the study presents some interesting points. Moreover, the topic faced in this study represents an important aspect that especially among adolescents and young people affected by cystinosis remains a serious limitation of cysteamine therapy, leading to an imparied effectiveness of this therapeutic strategy. 

However, I have some concerns that in my opinion should be addressed:

-I'm aware that measuring intraleukocytes cystine levels may be quite challenging and that few biochemical laboratories are able to carry out this analysis. However, this assay represents the analytical gold standard for the evaluation of adherence to cysteamine therapy. In fact, this approach would overcome all the limitations of self-reported measures and surveys that could be significantly affected by wrong information collected by each patient. Therefore, if the authors are not able to include this laboratory result, they should at least mention to the importance of performing this analysis in the Discussion section and in their Conclusions. In fact, future studies aimed to explore the factors influencing the poor-adherence should include also this methodological approach. 

- The authors should include a clinical evaluation of the patients involved in this study. A part from the Health Survey, were there clinical complications of cystinosis in these patients? How did this sub-optimal adherence impact on the clinical manifestations of the disease? I think that more follow-up data should be included by reporting not only the main complications observed due to the poor adherence but also the medical interventations proposed. This aspect in my opinion could increase the clinical interest of the study.

Minor points:

The Introduction should be shortened. In my opinion the studies referenced in this section could be moved in the Discussion section.         

Author Response

Please see the attachment, thank you!
